# Clinical Validation of the SECONDs Tool for Evaluating Disorders of Consciousness in Argentina

**DOI:** 10.3390/neurosci6040100

**Published:** 2025-10-07

**Authors:** María Julieta Russo, María de la Paz Sampayo, Paula Arias, Vanina García, Yanina Gambero, Mariano Maiarú, Florencia Deschle, Hernán Pavón

**Affiliations:** 1Dirección Docencia e Investigación, Santa Catalina Neurorehabilitación Clínica, Buenos Aires C1060AAF, Argentina; 2Instituto de Neurociencias (INEU) Fleni Consejo Nacional de Investigaciones en Científicas y Técnicas (CONICET), Buenos Aires C1060AAF, Argentina; 3Servicio de Kinesiología Motora, Santa Catalina Neurorehabilitación Clínica, Buenos Aires C1096, Argentina; 4Servicio de Terapia Ocupacional, Santa Catalina Neurorehabilitación Clínica, Buenos Aires C1096, Argentina; 5Servicio de Fonoaudiología, Santa Catalina Neurorehabilitación Clínica, Buenos Aires C1096, Argentina; 6Servicio de Neurociencias, Santa Catalina Neurorehabilitación Clínica, Buenos Aires C1096, Argentina

**Keywords:** assessment, coma, diagnosis, disorders of consciousness, validation study

## Abstract

Background: The Coma Recovery Scale–Revised (CRS-R) is the gold standard for diagnosing chronic disorders of consciousness (DoC); however, its clinical utility is limited by lengthy administration and the need for specialized training. The Simplified Evaluation of Disorders of Consciousness (SECONDs) provides a faster and more user-friendly alternative. Objective: This study aims to evaluate the validity and reliability of the Argentine adaptation of the SECONDs scale in adults with chronic DoC due to acquired brain injury. Methods: Twenty-nine patients were evaluated over two consecutive days by three blinded raters. On day one, rater A administered the SECONDs (A1) and rater B administered the CRS-R (B) to assess concurrent validity. On day two, rater A repeated the SECONDs (A2), and rater C performed an additional SECONDs assessment (C), permitting evaluation of intra-rater (A1 vs. A2) and inter-rater (A vs. C) reliability. Results: The SECONDs demonstrated excellent intra-rater (ICC = 0.98) and inter-rater (ICC = 0.86) reliability. Concurrent validity with the CRS-R was strong (r = 0.73, *p* < 0.001). Diagnostic agreement was high between A1 and B (κ = 0.75) and between both A1-A2 and A1-C (κ = 0.82). The median administration time was significantly shorter for the SECONDs (10 vs. 15 min; *p* < 0.001). Conclusion: The Argentine SECONDs is a valid, reliable, and efficient tool for the clinical assessment of DoC patients in rehabilitation settings.

## 1. Introduction

Detecting consciousness remains one of the most challenging tasks in clinical practice, and there is still no universally accepted, objective definition of this phenomenon. Because consciousness cannot be directly observed, clinical assessment primarily relies on behavioral observations and inferences regarding the individual’s underlying conscious state, especially in patients with disorders of consciousness (DoC).

Disorders of consciousness comprise a spectrum of clinical conditions resulting from severe brain injury, including coma, the vegetative state/unresponsive wakefulness syndrome (VS/UWS), the minimally conscious state (MCS), and emergence from the minimally conscious state (EMCS). Coma is defined by the absence of both wakefulness and awareness, with patients exhibiting no eye opening or purposeful behavior [1]. In contrast, VS/UWS is characterized by the recovery of arousal and sleep–wake cycles, but without any behavioral evidence of awareness of self or environment [2]. MCS is diagnosed when patients demonstrate clear, albeit inconsistent, signs of non-reflexive, purposeful behavior mediated by cortical activity, such as following commands, visual pursuit, or communication attempts [3]. Accurate diagnosis of these states relies on structured neurobehavioral assessment, typically using internationally validated criteria and standardized scales, most notably the Coma Recovery Scale–Revised (CRS-R) [4].

Consciousness itself is typically understood as comprising two key components: wakefulness, defined by spontaneous eye opening and arousal, and awareness, understood as the ability to respond coherently to both internal and external stimuli. However, clinical evaluations frequently fall short in precisely identifying consciousness. Differentiating among various DoC categories—such as coma, VS/UWS, and MCS—as well as ruling out other neurological conditions like locked-in syndrome or akinetic mutism, is complex and prone to error. The diagnostic uncertainty contributes to significant rates of misdiagnosis, which in turn impacts treatment decisions, the allocation of resources, and counseling regarding prognosis [5,6,7,8]. Diagnostic accuracy can be significantly improved by using standardized scales [5,7,8], conducting serial assessments at different times of day to account for fluctuations in consciousness [9], and integrating complementary functional studies when appropriate [10,11,12,13,14].

Since diagnosis drives rehabilitation strategies and is strongly correlated with outcomes, a precise assessment is essential. Several tools are used in both clinical and research settings to evaluate consciousness after brain injury, with the CRS-R being the most extensively validated and widely applied instrument in the last two decades due its robust psychometric properties [15].

Nevertheless, the CRS-R has certain limitations [16]. It is time-consuming, requiring observation and multiple assessments for an accurate diagnosis. It also requires specific training and experience. Moreover, while it provides a total score, this should not be used in isolation for diagnostic purposes.

To address these challenges, the Simplified Evaluation of CONsciousness Disorders (SECONDs) scale has recently been developed as a brief assessment tool evaluating patients with severe brain injury [16,17,18,19]. Notably, targeted assessment of the five most informative CRS-R items—fixation, visual pursuit, reproducible movement to command, automatic motor response, and localization to noxious stimulation—can identify the vast majority of MCS patients [20]. Based on these findings, the SECONDs scale was created and has already been validated in French [16], Spanish [19], Mandarin [18], and Italian [21].

The objective of this study was to assess the validity of the Argentine adaptation of the SECONDs scale in adults with chronic DoC resulting from acquired brain injury by examining: (1) its concurrent validity against the widely accepted diagnostic standard for DoC; (2) its internal consistency as well as inter-rater and intra-rater reliability; (3) its clinical utility, evaluated through administration time and diagnostic agreement in identifying DoC.

## 2. Materials and Methods

### 2.1. Formal Aspects

This study received approval from the Institutional Ethical Committee. Informed consent was obtained from the legal representatives of all subjects with DoC, following a thorough explanation of the study procedures. All designated relatives were monolingual Rioplatense Spanish. The research was conducted in accordance with the principles outlined in the Declaration of Helsinki (1975). The study’s design and reporting adhered to the STARD guidelines for studies of diagnostic accuracy [22].

### 2.2. Sampling and Study Design

This cross-sectional psychometric validation study was carried out between April and June 2024. We enrolled 29 DoC patients, 19 UWS/VS, 8 MCS, and 2 emergences from MCS (EMCS) hosted in the Santa Catalina Clinic in a dedicated unit for long-term brain injury care.

Inclusion criteria were as follows (1): age of 18 years or older, (2): impaired consciousness as determined by the attending physician based on CRS-R criteria for UWS, MCS, or EMCS (see Table 1), (3): at least 4 weeks post traumatic or non-traumatic brain injury resulting in coma, defined by a Glasgow Coma Scale score below 8 at the time of acute admission, With no upper limit set for time since injury to better capture the clinical heterogeneity of this population; (4): native Spanish speakers.

Exclusion criteria included diagnosed with coma, including irreversible coma or brain death; those who had received neuromuscular blocking agents or sedative medications within 24 h prior to clinical evaluation; individuals with injuries affecting both eyes, tympanic membranes, or inner ears, patients with severe trauma to both upper or lower limbs; or those in unstable physical condition (e.g., absence of mechanical ventilation, ongoing sedation, active infection, or seizures).

### 2.3. Assessment

We designed a standardized neurobehavioral protocol specifically for patients with chronic DoC within the rehabilitation setting. The clinical evaluation aims to assess the patient’s level of alertness in response to verbal and painful stimuli, establish the severity of consciousness impairment, and inform ongoing monitoring and therapeutic decision-making. Based on international recommendations, the clinical protocol aims to (1) rule out all those confounding clinical factors; (2) use a standardized and validated scale such as the CRS-R [4] and SECONDs [16]; (3) avoid making a diagnosis only with the first evaluation, trying as far as possible to plan at least 5 evaluation sessions in the term of 2 weeks; (4) review the pharmacological scheme that allows planning a gradual and progressive decrease in drugs that alter neuroplasticity and the incorporation of drugs that stimulate consciousness in adequate doses; (5) request the necessary complementary methods to improve the precision of the clinical interpretation and to exclude other causes; (6) if available, functional studies (MRI and/or EEG) can be requested when there is suspicion of covert consciousness.

### 2.4. Measures

#### 2.4.1. Coma Recovery Scale-Revised (CRS-R)

The CRS-R [4] is widely recognized as the gold standard behavioral assessment tool for individuals with Disorders of Consciousness (DoC). This standardized scale consists of 23 hierarchically arranged items across six subscales: auditory, visual, motor, oromotor/verbal, communication, and arousal. It is specifically designed to differentiate between coma, VS/UWS, MCS, and EMCS. These subscales included hierarchically organized items that reflect behaviors mediated by brainstem, subcortical, and cortical regions: auditory, visual, motor, oromotor/verbal, communication, and arousal functions. The total CRS-R score is obtained by summing the six subscale scores, ranging from 0 to 23.

Within each subscale, the lowest scoring items correspond to reflexive responses, while the scores indicate cognitively mediated behaviors. Scoring follows standardized criteria based on the presence or absence of clearly defined behavioral responses. Additionally, the scoring form links specific behaviors to diagnostic categories, facilitating clinical interpretation. The CRS-R has demonstrated strong psychometric properties, including high inter-rater reliability (e.g., κ = 0.92 in Japanese validation, ICC = 0.719 in Chinese, κ = 0.99 in Russian) [23,24,25], good to excellent test–retest reliability (e.g., ρ = 0.92 in Japanese, ICC = 0.87 in Chinese, r = 0.997 in Russian) [23,24,25], and high internal consistency (e.g., Cronbach’s α = 0.91 in Japanese, 0.84 in Chinese, 0.87 in Russian) [23,24,25]. Its excellent content validity covers all criteria defined by the Aspen Workgroup [4]. The CRS-R also exhibits good concurrent validity with other assessment tools, such as the Glasgow Coma Scale (GCS) and Disability Rating Scale (DRS), and demonstrates higher diagnostic sensitivity in detecting subtle signs of consciousness compared to scales like the GCS and FOUR score [23,24,25]. While administration typically takes between 25 and 40 min, repeated serial assessments (e.g., at least five within two weeks) are crucial to account for fluctuations in a patient’s level of consciousness and to reduce potential misdiagnosis rates, which can be substantial (30–40%) when relying on single evaluations or clinical consensus alone [9,26,27]. Examiner training and experience are also noted to influence its reliability [4]. The measurement properties of a Spanish version of the CRS-R showed similar results [28].

#### 2.4.2. Simplified Evaluation of CONsciousness Disorders (SECONDs)

The SECONDs scale [12,13] is composed of eight items arranged in order of increasing complexity, adapted from the CRS-R: arousal response (scored 1), localization to pain (2), visual fixation (3), visual pursuit (4), oriented behaviors (5), command-following (6), intentional communication (scored 7), and functional communication (8). The total score corresponds to a diagnostic category, ranging from 0 (coma), 1 (UWS), 2–5 (MCS−), 6–7 (MCS+) to 8 (EMCS).

Additionally, the authors propose an index score calculated by summing the points earned across all items assessed, with a possible range from 0 to 100. The SECONDs scale has demonstrated to be a rapid, reliable and user-friendly tool for diagnosing DoC.

### 2.5. Translation

With the original author’s permission [16,17], the SECONDs scale was translated into Spanish following a rigorous process based on international guidelines for instrument adaptation [19]. The original English, which includes detailed administration guidelines to ensure standardized and reproducible use [17], was first translated into standard Spanish by a native Spanish speaking neurologist familiar with DoC. Next, an independent, certified bilingual translator performed a back-translation from Spanish to English. Both versions—Spanish [19] and Argentine—and the back-translation were then reviewed and compared by an expert committee of clinicians, neuropsychologists, and language professionals to resolve discrepancies and ensure conceptual equivalence. Two rounds of iterative revisions were conducted before approval of the final Argentine Spanish version.

Notably, several phrases and instructions were modified to align with Rioplatense Spanish, the regional variant spoken in Argentina, focusing on terminology and expressions familiar within local clinical practice. For example, the instruction “fije la mirada” in the standard Spanish version was rendered as “mire fijamente” or “mantenga la vista fija” in the Argentine version to ensure greater clarity for local users. The command “mueva la mano” (“move your hand”) was favored over alternatives such as “muévase la mano,” which could cause confusion. In the autobiographical questions used for communication testing, the wording “¿Tiene hijos?” (“Do you have children?”) was preferred over “¿Tiene niños?” to avoid ambiguity in the Argentine context. Everyday objects and sample commands were also revised to reflect items commonly recognized in Argentina’s hospitals and rehabilitation settings. Through these lexical and syntactic adaptations, the final version of the scale achieves both linguistic accuracy and cultural appropriateness for Argentine Spanish-speaking populations, increasing its clinical relevance and feasibility of administration.

All back-translation versions (https://zenodo.org/records/13946392) (accessed on 26 September 2025) and the Spanish translation (https://zenodo.org/records/13946488) (accessed on 26 September 2025) has been registered in the Zenodo repository and are freely accessible to the public.

### 2.6. Procedure

Throughout the evaluation period, all patients continued to receive standard medical treatment and rehabilitation. The study involved three physical therapists (Raters A, B, and C), all of whom were new to using the SECONDs scale. Prior to data collection, the raters underwent training on the administration of the SECONDs following the published guidelines [17]. Patients were diagnosed with UWS, MCS, or EMCS based on CRS-R assessments according to the Aspen Workgroup criteria, by a senior neurologist (see Table 1).

On the first day, Rater A performed the initial SECONDs assessment (A1), while Rater B conducted the CRS-R evaluation on the same day. To minimize fluctuations in patients’ responsiveness and avoid fatigue, a 40- to 60-min interval was maintained between the two assessments. The order of these assessments (A1 and B) was randomized. Although both raters visited the patients simultaneously, only one conducted the assessment while the other waited outside, preventing observation or knowledge of each other’s results; assessments were submitted independently.

On the following day, Rater A repeated the SECONDs assessment (A2), and Rater C administered de SECONDs once. The same time interval was maintained between these two evaluations (see Figure 1). All raters applied diagnostic criteria consistent with senior neurologists to classify patients’ levels of consciousness impairment as UWS, MCS, or EMCS.

### 2.7. Statistical Analysis

Data analyses were performed using SPSS statistics (version 22, IBM Corp., Chicago, IL, USA). Normally distributed continuous variables are presents as mean with standard deviations, while non-normally distributed variables are reported as medians with interquartile ranges. Categorical data are summarized using counts and prevalence rates.

To assess concurrent validity, Spearman’s partial correlation was used to examine relationships between the SECONDs subscales, the additional index and the total SECONDs score compared to the total CRS-R score. Internal consistency of the SECONDs was evaluated using Cronbach’s alpha coefficient and inter-subscale correlation coefficients.

Inter-rater and intra-rater reliability were estimated through intra-class correlation coefficients (ICC), with values between 0.60 and 0.74 indicating good clinical significance, and values ≥0.75 considered excellent. Agreement between diagnoses made by CRS-R and SECONDs raters was assessed with weighted Kappa coefficients.

The Mann–Whitney U test was employed to compare administration times between the SECONDs and the CRS-R. Statistical significance was set at *p* < 0.05 with a 95% confidence interval for all tests.

## 3. Results

A total of 29 individuals with DoC were evaluated over two consecutive days by three independent, blinded raters. On one day, each participant underwent both a CRS- R and a SECONDs assessment, while on the other day, two SECONDs assessments were administered. The mean age of the cohort was 55.48 years (SD = 16.28), with 16 participants (55%) being male. Regarding etiology, 5 individuals (17%) had experienced a stroke, 9 (31%) had sustained traumatic brain injury, and 15 (52%) had an anoxic brain injury. The median time since brain injury was 20 months (interquartile range: 7.50–59.00). Table 2 depicts the characteristics of this study’s participants.

### 3.1. Concurrent Validity

Table 3 presents the correlation coefficients between the SECONDs scale and the CRS-R. The Spearman correlation was significant for the total scores of the SECONDs and CRS-R (r = 0.73; *p* < 0.001), as well as for the additional index score of the SECONDs (r = 0.81, *p* < 0.001). Each subscale of the SECONDs additional index also showed significant associations with the CRS-R, with correlation coefficients ranging from 0.43 to 0.73 (all *p*  <  0.05).

### 3.2. Internal Consistency

The Cronbach’s alpha coefficient for the additional index score of the SECONDs scale was 0.77, indicating a reasonably homogeneous measure of neurobehavioral function in patients with DoC. Removing any individual subscale had minimal impact on the overall Cronbach’s alpha. Correlations among the subscales (see Table 4) were generally satisfactory for most items, except for the Arousal and Pain localization subscales, which showed weaker correlations (ICC not statistically significant). The strongest interrelationships were observed among the Oriented behaviors, Visual pursuit, and Visual fixation subscales.

### 3.3. Reliability

Table 5 summarizes the intra-rater and inter-rater correlation coefficients (ICCs) for both the total score and the additional index score of the SECONDs scale. Intra-rater reliability was nearly perfect (ICC = 0.98; *p* < 0.001), demonstrating excellent consistency when assessments were repeated by the same evaluator. All ICC values were statistically significant except for the Pain Localization subscale.

Inter-rater reliability for the total SECONDs score was also high (ICC = 0.86; *p* < 0.001), indicating the diagnostic accuracy remained robust across different raters. However, the Pain Localization and Arousal subscales showed lower inter-rater reliability and did not reach statistical significance.

### 3.4. Clinical Utility

The median administration time for the SECONDs scale was significantly shorter than that of the CSR-R (10 min, IQR 9–12 versus 15 min, IQR 10–16; U = 172.5, *p* < 0.001).

In this study, 29 participants were assessed using the CRS-R, yielding diagnoses of EMCS in 2 patients, MCS in 8, and UWS in 18. Diagnostic agreement between raters was high: 25 out of 29 cases (κ = 0.75; *p* < 0.001) between Raters A1 and B, 26 out of 29 (κ = 0.82; *p <* 0.001) between A1 and A2, and 26 out of 29 (κ = 0.82; *p* < 0.001) between A1 and C.

When distinguishing between MCS− and MCS+ categories (see Figure 2), diagnostic disagreement between the SECONDs and CRS-R scales occurred in 5 of 29 participants (17%). Among these, 3 participants received a higher level of awareness diagnosis with the SECONDs compared to the CRS-R, while 2 participants were better classified by the CRS-R. These discrepancies primary involved differences in detecting visual pursuit (observed only with SECONDs in 1 participant and only with CRS-R in another) and command-following (detected only with SECONDs in 2 participants and only with CRS-R in 1 participant).

## 4. Discussion

This study constitutes the first validation of the Argentine adaptation of the SECONDs scale in a cohort of patients with DoC secondary to acquired brain injury. Our findings demonstrate that the SECONDs scale is a valid, reliable, and efficient tool for assessing of consciousness in this population.

We found strong concurrent validity with the established CRS-R scale, demonstrated by significant correlations between the total scores of both instruments, which increased further when including the additional SECONDs index. These findings align with previous research [16,18,19,21], confirming that the Argentine version of the SECONDs scale effectively measures levels of consciousness.

The Cronbach’s alpha coefficient supports good structure validity for the additional index score. Although the Arousal and Pain Localization subscales showed weaker correlations, their exclusion did not affect the scale’s overall internal consistency. The Pain Localization subscale is conditional and scores zero on the additional index when command-following is present; fluctuations in arousal can significantly impact behavioral responsiveness. While arousal changes naturally influence total SECONDs scores, this study did not specifically examine this effect. The additional index aims to capture subtle clinical changes over repeated assessments without altering the overall diagnosis [17]. Thus, evaluating arousal and pain localization remains essential to reflect DoC diagnoses accurately.

Our results align with previous validation studies. For instance, Aubinet et al. [16] reported excellent inter-rater reliability (ICC = 0.93) and strong concurrent validity with the CRS-R (r = 0.80), findings that are comparable to those observed in our study (inter-rater ICC = 0.86; concurrent validity r = 0.73). Similarly, the Mandarin [18], Spanish [19], and Italian [21] versions showed excellent reliability and validity, supporting the scale’s robustness across languages and cultures. Our study confirms adequate reliability for both the total and additional index scores, demonstrating stability across raters and repeated measures.

Regarding diagnostic agreement, our kappa values (κ = 0.75 to 0.82) are in line with the original validation (κ = 0.84) [16], Spanish version (κ = 0.87) [19], and Italian version (κ = 0.83) [21], reinforcing its capacity to distinguish consciousness reliably. Discrepancies with the CRS-R mainly involved behaviors assessed differently, particularly in visual pursuit and command-following subscales. Variations in assessment conditions and patient variability likely contributed [16].

A practical advantage of the SECONDs scales is its shorter administration time compared to the CRS-R (median 10 min vs. 15 min; U = 172.5, *p* < 0.001), facilitating its use in settings with limited staff or time constraints [16,21].

### 4.1. Clinical Implications

The Argentine version of the SECONDs scale provides valuable advantages for evaluating patients with prolonged DoC, particularly in low- and middle-income countries where resources and specialized training are often limited. Its brevity and ease of use enable more frequent monitoring, potentially leading to earlier detection of clinical changes and timely rehabilitation adjustments. Strong concordance with CRS-R supports its use as an alternative or complementary tool. The validated Spanish version fills a critical gap, as most DoC tools are in English or other major languages.

### 4.2. Study Limitations

The small sample size (n = 29) may limit generalizability. Larger multicenter studies are needed to confirm psychometric properties across diverse populations. Participants came from a single specialized rehabilitation center, possibly introducing selection bias. Raters were physical therapists trained in SECONDs; performance by other professionals or in less controlled settings remains to be evaluated.

This study did not assess sensitivity to subtle changes over time or intervention response; future research should explore longitudinal utility.

### 4.3. Future Directions

Further research should aim to evaluate the performance of the SECONDs scale in larger, more heterogeneous samples and across diverse Spanish-speaking countries, strengthening the external validity and cross-cultural applicability of the tool. Beyond multicenter validation, it is important to assess the instrument’s responsiveness to subtle clinical changes over time and its sensitivity to interventions aimed at improving consciousness. Longitudinal studies tracking SECONDs scores alongside patient outcomes could better establish its predictive value.

At the tool level, several opportunities for improvement merit exploration. For example, the psychometric properties of the Arousal and Pain Localization subscales could be refined, potentially through item revision or more detailed operational definitions to enhance inter-rater reliability. Additionally, optimizing the assessment guidelines—such as standardized stimuli or clearer instructions—may reduce variability among evaluators and improve reproducibility. The development of digital or app-based SECONDs versions could facilitate training, standardize scoring, and allow for real-time data capture in a variety of clinical contexts. Another important direction is to harmonize the tool’s diagnostic thresholds and scoring rules with other behavioral and neurophysiological measures, fostering truly multimodal assessment protocols for disorders of consciousness.

Ultimately, continued iterative refinement of the SECONDs scale, supported by both psychometric analyses and direct clinician feedback, will help maximize its clinical utility and ensure its relevance across settings and patient populations.

## 5. Conclusions

This study confirms the Argentine SECONDs scale as a valid, reliable, and efficient tool for evaluating prolonged DoC. Its integration into neurobehavioral assessments may improve diagnostic accuracy and patient care, particularly in resource-limited settings. Ongoing research will further elucidate its added value in comprehensive evaluation and management.

## Figures and Tables

**Figure 1 neurosci-06-00100-f001:**
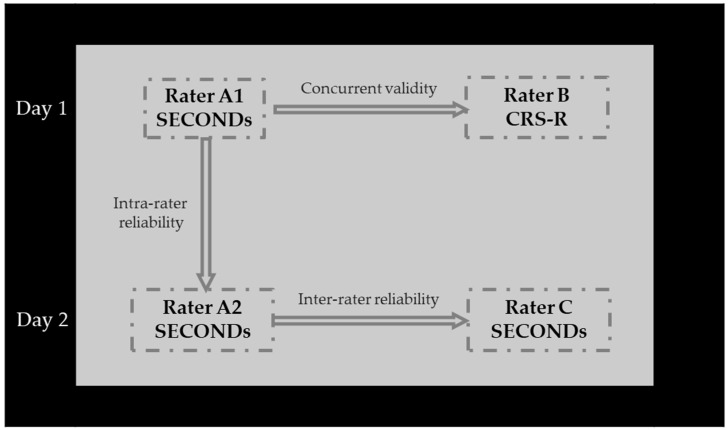
Research procedure. CRS-R: Coma Recovery Scale-Revised; SECONDs: Simplified Evaluation of CONsciousness Disorders.

**Figure 2 neurosci-06-00100-f002:**
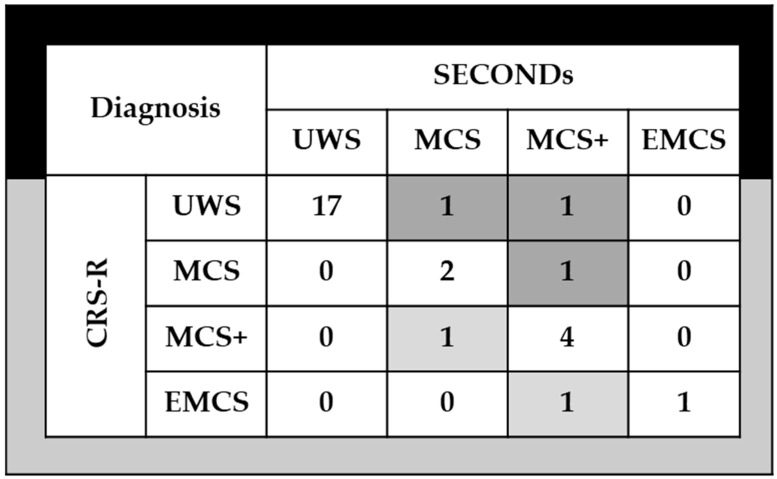
Diagnostic agreement between the SECONDs and the CRS-R. Numbers in the cells indicate the number of patients with matching diagnoses following SECONDs and CRS-R. Shaded boxes indicate differential diagnoses: dark gray when the SECONDs result was better than the CRS-R, and light gray when the CRS-R result was better than the SECONDs. CRS-R: Coma Recovery Scale-Revised; SECONDs: Simplified Evaluation of CONsciousness Disorders; UWS: Unresponsive Wakefulness Syndrome; MCS minimally conscious state; EMCS: emergence from minimally conscious state.

**Table 1 neurosci-06-00100-t001:** Diagnostic criteria for disorders of consciousness by CRS-R.

Scale	UWS	MCS	EMCS
CRS-R	Auditory ≦ 2 AND Visual ≦ 1 AND Motor ≦ 2 AND Oromotor/Verbal ≦ 2 AND Communication = 0 AND Arousal ≦ 2	Auditory = 3–4 OR Visual = 2–5 OR Motor = 3–5 OR Oromotor/Verbal = 3 OR Communication = 1	Motor = 6 Communication = 2

CRS-R: Coma Recovery Scale-Revised; UWS: Unresponsive Wakefulness Syndrome; MCS minimally conscious state; EMCS: emergence from minimally conscious state.

**Table 2 neurosci-06-00100-t002:** Characteristics of this study’s participants.

Variable	Value
Age, yr. (mean ± standard deviation)	55.48 (16.28)
Gender distribution, male/female	16/13
Time post-injury, months (median [interquartile range])	20 (7.50–59.00)
CRS-R	
Total Score (median [interquartile range])	6 (4.00–11.00)
Duration of administration, minutes (median [interquartile range])	15 (10.00–16.00)
SECONDs, A1 rater	
Total Score (median [interquartile range])	1 (1.00–5.50)
Additional Index Score (median [interquartile range])	4 (3.00–25.00)
Duration of administration, minutes (median [interquartile range])	10 (9.00–12.00)
SECONDs, A2 rater	
Total Score (median [interquartile range])	1 (1.00–5.50)
Additional Index Score (median [interquartile range])	4 (4.00–32.50)
Duration of administration, minutes (median [interquartile range])	10 (9.00–10.15)
SECONDs, C rater	
Total Score (median [interquartile range])	1 (1.00–6.00)
Additional Index Score (median [interquartile range])	4 (3.50–33.50)
Duration of administration, minutes (median [interquartile range])	10 (9.00–12.50)
Etiology	
*Anoxia*	15 (52%)
*Trauma*	9 (31%)
*Stroke*	5 (17%)
Clinical state according to Aspen Criteria	
*UWS*	19 (65.52%)
*MCS*	8 (27.58%)
*EMCS*	2 (6.90%)

CRS-R: Coma Recovery Scale-Revised; SECONDs: Simplified Evaluation of CONsciousness Disorders; UWS: Unresponsive Wakefulness Syndrome; MCS minimally conscious state; EMCS: emergence from minimally conscious state.

**Table 3 neurosci-06-00100-t003:** Correlation between SECONDs and CRS-R scores.

SECONDs Subscales	CRS-R
Communication	0.45 *
Command-following	0.51 **
Oriented behaviors	0.73 **
Visual pursuit	0.42 *
Visual fixation	0.60 **
Pain localization	0.43 *
Arousal	0.64 **
Total SECONDs score	0.73 **
Additional index SECONDs score	0.81 **

CRS-R: Coma Recovery Scale-Revised; SECONDs: Simplified Evaluation of CONsciousness Disorders. * *p* < 0.05; ** *p* < 0.001.

**Table 4 neurosci-06-00100-t004:** Correlations among the subscales of the additional index SECONDs score.

SECONDs Subscales	Total SECONDs Score	Communication	Command- Following	Oriented Behaviors	Visual Pursuit	Visual Fixation	Pain Localization	Arousal
Total SECONDs score	-							
Communication	0.57 **	-						
Command-following	0.79 **	0.32 *	-					
Oriented behaviors	0.73 **	0.29 *	0.66 **	-				
Visual pursuit	0.89 **	0.41 *	0.52 **	0.53 **	-			
Visual fixation	0.90 **	0.42 *	0.54 **	0.56 **	0.92 **	-		
Pain localization	0.16	−0.05	−0.11	−0.07	0.32	0.36 *	-	
Arousal	0.34 *	0.14	0.10	0.10	0.38 *	0.38 *	0.11	-

SECONDs: Simplified Evaluation of CONsciousness Disorders. * *p* < 0.05; ** *p* < 0.001.

**Table 5 neurosci-06-00100-t005:** Intra-class coefficients (ICCs) of total score of the SECONDs between raters (inter-rater and intra-rater reliability).

	CCI	IC95%		F	*p*-Value
		Lower Limit	Upper Limit		
Intra-rater reliability					
Communication	0.78	0.57	0.88	7.702	<0.0001
Command-following	0.76	0.55	0.88	7.269	<0.0001
Oriented behaviors	0.93	0.86	0.97	27.465	<0.0001
Visual pursuit	0.80	0.62	0.92	9.306	<0.0001
Visual fixation	0.98	0.95	0.98	81.500	<0.0001
Arousal	0.67	0.40	0.83	4.927	<0.0001
Total SECONDs score	0.98	0.96	0.99	109.299	<0.0001
Additional index SECONDs score	0.94	0.88	0.97	34.277	<0.0001
Inter-rater reliability					
Communication	0.46	0.12	0.70	2.697	0.0005
Command-following	0.77	0.57	0.89	8.171	<0.0001
Oriented behaviors	0.78	0.58	0.89	8.888	<0.0001
Visual pursuit	0.75	0.50	0.88	8.132	<0.0001
Visual fixation	0.88	0.77	0.94	16.360	<0.0001
Total SECONDs score	0.86	0.72	0.93	12.850	<0.0001
Additional index SECONDs score	0.96	0.92	0.98	50.815	<0.0001

ICC: Intra-class coefficients; SECONDs: Simplified Evaluation of CONsciousness Disorders.

## Data Availability

The data that support the findings of this study are available from the corresponding author upon reasonable request.

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
