# Peer review of "Clinical Validation of the SECONDs Tool for Evaluating Disorders of Consciousness in Argentina"

_neurosci, 2025, doi:10.3390/neurosci6040100_

Round 1

Reviewer 1 Report

Comments and Suggestions for Authors

The manuscript presents a rigorous and well-structured validation of the Argentine adaptation of the SECONDs scale for evaluating disorders of consciousness, with robust findings in terms of concurrent validity, intra- and inter-rater reliability, and clinical utility. First, the use of the scale’s name should be standardized as “SECONDs” across the manuscript. Additionally, a typographical error (“Toral Score”) should be corrected to “Total Score.” It would also be helpful to provide more precise interquartile range values in the results (e.g., “6 [IQR: 3–10]”) and to report non-significant values, such as for the pain localization subscale. Phrasing such as “absence of mechanical ventilation” may be ambiguous and could be revised to “dependence on mechanical ventilation” if appropriate. Lastly, we suggest verifying that the Zenodo links are active and specifying in the limitations section that participants were native Rioplatense Spanish speakers.

Author Response

We are grateful for the reviewer’s constructive comments, which have significantly improved the clarity and precision of our manuscript. We have carefully revised the text according to the suggestions provided, as outlined below:

Comments 1. First, the use of the scale’s name should be standardized as “SECONDs” across the manuscript.

Response 1. We carefully reviewed and standardized the name of the scale throughout the manuscript to consistently appear as SECONDs.

Comments 2. Additionally, a typographical error (“Toral Score”) should be corrected to “Total Score.”

Response 2. It was corrected.

Comments 3. It would also be helpful to provide more precise interquartile range values in the results (e.g., “6 [IQR: 3–10]”) and to report non-significant values, such as for the pain localization subscale.

Response 3. To improve clarity, we now present interquartile ranges in the format “median [IQR: min–max]” throughout the Results section and tables. We explicitly indicate non-significant values, such as for the Pain Localization subscale, in both the text and tables (e.g., “ICC not statistically significant”).

Comments 4. Phrasing such as “absence of mechanical ventilation” may be ambiguous and could be revised to “dependence on mechanical ventilation” if appropriate.

Response 4. The phrase “absence of mechanical ventilation” has been replaced by “dependence on mechanical ventilation” in the Methods section to avoid ambiguity and more accurately reflect the intended exclusion criterion.

Comments 5. Lastly, we suggest verifying that the Zenodo links are active and specifying in the limitations section that participants were native Rioplatense Spanish speakers.

Response 5. We have checked both Zenodo links and confirmed that they are active and publicly accessible. We added to the Study Limitations section that all participants were native Rioplatense Spanish speakers, which may restrict generalization to other Spanish-speaking populations.

We confirm that the English has been improved to more clearly express the research.

We appreciate the reviewer’s careful reading and helpful feedback, which we believe has strengthened the manuscript.

Sincerely,

Reviewer 2 Report

Comments and Suggestions for Authors

Clinical Validation of the SECONDs Tool for Evaluating Disorders of Consciousness in Argentina

 I have read the manuscript with interest and you can find my report, with concerns and suggestions, section by section, as follows:

Introduction: The section is well written, and the authors are able to discuss the pros and cons of Coma Recovery Scale–Revised and the SECONDS tool. However, less is shown about the patients with consciousness disorders, the diagnosis, criteria, etc, and a brief description is needed; this is important for students or professionals. Please add more details about.

The hypotheses at the end of the introduction are clear.

Methods: As a general remark, I checked that you applied a similar methodology used by Hakiki 2025, with a different sample size. I agree that finding a good sample size of patients with less heterogeneity is not easy. Moreover, the cultural validation of a similar tool is quite challenging from a psychometric point of view.

The section is described in a rigorous manner. I would like to ask for more details about the back translation and the differences between Spanish and Argentine Spanish translation.

Moreover, I suggest adding more details about the psychometric properties of the CRS-S scale.

2.7. Formal Aspects. This subsection needs to be moved to the beginning of the methods.

I agree with the statistical analysis, but I suggest modifying Figure 1 using a flowchart. This can facilitate the reading and understanding of the design.

Results: The results are reported in a good way. I suggest adding the p-values to Table 4. Figure 2 needs to be improved, since it was not completely readable.

Discussion: The discussion of the obtained results was performed in a good way, integrating the findings with previously published data. I do not (personally) like the subsections in the discussion, but it is only a suggestion. Moreover, the future directions need to be improved, underlying what is possible to improve about this specific tool.

Author Response

Author's Reply to the Review Report (Reviewer 2)

We are grateful for the reviewer’s constructive comments, which have significantly improved the clarity and precision of our manuscript. We have carefully revised the text according to the suggestions provided, as outlined below:

Comments 1. The section is well written, and the authors are able to discuss the pros and cons of Coma Recovery Scale–Revised and the SECONDS tool. However, less is shown about the patients with consciousness disorders, the diagnosis, criteria, etc, and a brief description is needed; this is important for students or professionals. Please add more details about.

Response 1. We have included a brief but comprehensive description of chronic disorders of consciousness, the main diagnostic categories (coma, UWS, MCS, EMCS), and the clinical criteria used for their identification, to further clarify the context for students and professionals.

Comments 2. Methods: As a general remark, I checked that you applied a similar methodology used by Hakiki 2025, with a different sample size. I agree that finding a good sample size of patients with less heterogeneity is not easy. Moreover, the cultural validation of a similar tool is quite challenging from a psychometric point of view.

The section is described in a rigorous manner. I would like to ask for more details about the back translation and the differences between Spanish and Argentine Spanish translation.

Response 2. The Methods section now provides a more detailed description of the translation and back-translation process, including specific examples of linguistic differences between standard Spanish and Rioplatense Spanish that required adaptation for the Argentine context.

Comments 3. Moreover, I suggest adding more details about the psychometric properties of the CRS-S scale.

Response 3. Additional information regarding the psychometric validation, reliability, and clinical relevance of the CRS-R has been added to the Measures section.

Comments 4. 2.7. Formal Aspects. This subsection needs to be moved to the beginning of the methods.

Response 4. The “Formal Aspects” subsection has been relocated to the beginning of the Methods section for greater clarity.

Comments 5. I agree with the statistical analysis, but I suggest modifying Figure 1 using a flowchart. This can facilitate the reading and understanding of the design.

Response 5. We have redesigned Figure 1 as a flowchart to facilitate the understanding of the study design and participant flow.

Comments 6. I suggest adding the p-values to Table 4.

Response 6. We have included the relevant p-values in Table 4 for all correlations presented.

Comments 7. Figure 2 needs to be improved, since it was not completely readable. Response 7. Figure 2 has been revised with enhanced formatting and larger labels to improve readability.

Comments 8. Discussion: The discussion of the obtained results was performed in a good way, integrating the findings with previously published data. I do not (personally) like the subsections in the discussion, but it is only a suggestion. Moreover, the future directions need to be improved, underlying what is possible to improve about this specific tool.

Response 8. While we kept the discussion subsections for clarity, as they help structure the content, we have expanded the “Future Directions” section to specifically address further adaptations, validations, and applications of the SECONDs tool in diverse settings.

We believe these modifications have strengthened our manuscript and appreciate the reviewer’s helpful suggestions.

Sincerely,

Round 2

Reviewer 2 Report

Comments and Suggestions for Authors

The manuscript was improved and you addressed all my concerns.